# Cytokine Pattern of Peripheral Blood Mononuclear Cells Isolated from Children Affected by Generalized Epilepsy Treated with Different Protein Fractions of Meat Sources

**DOI:** 10.3390/nu14112243

**Published:** 2022-05-27

**Authors:** Maria Giovanna Cilberti, Antonella Santillo, Anna N. Polito, Giovanni Messina, Antonella della Malva, Mariangela Caroprese, Agostino Sevi, Marzia Albenzio

**Affiliations:** 1Department of Agriculture, Food, Natural Resources, and Engineering (DAFNE), University of Foggia, 71122 Foggia, Italy; maria.ciliberti@unifg.it (M.G.C.); antonella.dellamalva@unifg.it (A.d.M.); mariangela.caroprese@unifg.it (M.C.); agostino.sevi@unifg.it (A.S.); marzia.albenzio@unifg.it (M.A.); 2Complex Structure of Neuropsychiatry Childhood-Adolescence of Ospedali Riuniti of Foggia, Viale Pinto, 71122 Foggia, Italy; annanupolito@gmail.com; 3Department of Clinical and Experimental Medicine, University of Foggia, 71122 Foggia, Italy; giovanni.messina@unifg.it

**Keywords:** epilepsy, fish, meat, proteins, cytokines, inflammation

## Abstract

The objective of the present study was the evaluation of cytokine patterns in terms of TNF-α, IL-10, IL-6, and IL-1β secretion in peripheral blood mononuclear cell (PBMC) supernatants isolated from blood of children affected by generalized epilepsy and treated in vitro with myofibrillar, sarcoplasmic, and total protein fractions of meat and fish sources. Children with generalized epilepsy (EC group, *n* = 16) and children without any clinical signs of disease, representing a control group (CC group *n* = 16), were recruited at the Complex Structure of Neuropsychiatry Childhood-Adolescence of Policlinico Riuniti (Foggia, Italy). Myofibrillar (MYO), sarcoplasmic (SA), and total (TOT) protein fractions were obtained from *longissimus thoracis* muscle of beef (BF) and lamb (LA); from *pectoralis* muscle of chicken (CH); and from dorsal white muscle of sole (*Solea solea*, SO), European hake (*Merluccius merluccius*, EH), and sea bass fish (*Dicentrarchus labrax*, SB), respectively. PBMCs were isolated from peripheral blood of EC and CC groups, and an in vitro stimulation in the presence of 100 μg/mL for each protein fraction from different meat sources was performed. Data were classified according to three different levels of cytokines produced from the EC group relative to the CC group. TNF-α, IL-10, and IL-6 levels were not affected by different meat fractions and meat sources; on the contrary, IL-1β levels were found to be significantly affected by the tested proteins fractions, as well as different meat sources, in high-level cytokine group. On average, the protein fractions obtained from LB, BF, and CH meat sources showed a higher level of IL-1β than the protein fractions obtained from EH and SB fish samples. When all cytokine classes were analyzed, on average, a significant effect was observed for IL-10, IL-1β, and TNF-α. Data obtained in the present study evidence that the nutritional strategy based on protein from fish and meat sources may modulate the immunological cytokine pattern of infants with generalized epilepsy.

## 1. Introduction

Epilepsy represents a neurological disease with an incidence rate of 33/100,000–82/100,000 cases per year and affects the development of children, as well as their quality of life [1]. Epilepsy can be classified into focal epilepsy, generalized epilepsy, combined generalized, and unknown according to the standards of the International League against Epilepsy (ILAE). Several antiepileptic drugs (AEDs) have been recently used for epilepsy treatment; however, 20–30% of patients are found to be AED-resistant [2]. Due to the crucial role of diet in regulating inflammatory status and intestinal microbiota composition, nutritional strategies have gained interest in epilepsy research. A ketogenic diet is considered a valid therapeutic strategy to treat children with intractable epilepsy, reducing inflammation at the intestinal level and preserving the homeostasis of the intestinal microbiome [3]. Whereas the mechanisms of action of a ketogenic diet related to seizure control are still being actively investigated, many theories have been proposed in this regard. One possible mechanism of action of a ketogenic diet is the neuroprotective effect of reducing oxidative stress at the mitochondrial level via ketone and polyunsaturated fatty acid (PUFA) production [4]. However, a ketogenic diet, in relation to low-protein consumption, may result in limitation of the levels of essential ammino acids, which must be provided in the diet and include isoleucine, leucine, lysine, methionine, phenylalanine, threonine, tryptophan, and valine [5]. Histidine ammino acids are considered essential for the first year of life, and a deficiency of threonine, isoleucine, and histidine has been found to exacerbate seizures in rats [6]. Furthermore, the limitation of essential ammino acids can excite a zone of high epileptogenicity contained in the anterior piriform cortex of the brain [7]. Overall, nutritional factors can play an important role in the regulation of electrical activity in the brain; thus, a deficiency of thiamine, pyridoxine, vitamin D, calcium, magnesium, and carnitine may be a cause of seizures. A subnormal concentration of these nutrients has been observed in most patients with epilepsy [8]. 

The inflammatory processes exert a crucial role in facilitating the development of epilepsy, and, in turn, epileptic seizures can trigger an inflammatory response [9], as supported by experimental and clinical findings [10]. Cytokines are able to correlate the central nervous system (CNS) and immune system in a bidirectional mode [11], mediating the development and regulation of the inflammatory and immune responses. Proinflammatory cytokines have been found to increase after single generalized or prolonged seizures [12]; thus, both the interleukin-1 receptor antagonist (IL-1Ra) and IL-6 have demonstrated neuroprotective and anticonvulsive effects during status epilepticus [13,14]. Based on previous statements, the relevance of the role of peripheral blood mononuclear cells (PBMC) is related to recent research on the pathogenesis of epilepsy with the aim of exploring the mechanisms mediating peripheral-to-central cell infiltration in both human and mouse models [15]. 

The effect of diet on the modulation of the gut–brain axis, brain inflammatory reactions, and dietary allergic disorders in epilepsy have been recently reviewed in childhood epilepsy [16]. The effects of protein fractions on human health have gained attention for their important role in stimulating innate immune response and activating pro- and anti-inflammatory cytokine release at a systemic level [17]. Milk protein fraction is a very heterogeneous nutritional component that impacts human health, and the related milk genetic polymorphism could play a role in the level and type of protein and derived peptides able to influence the in vitro cytokine pattern in epileptic infants [18]. Therefore, it could be of interest to explore the immunomodulatory role of protein from different animal protein food sources. 

Our hypothesis is that protein fractions extracted from different meat and fish sources could have a potential immunomodulatory or stimulatory effect on cytokine patterns of infants with generalized epilepsy. The aim of this study was the investigation of TNF-α, IL-10, IL-6, and IL-1β cytokines secretion in PBMC supernatants isolated from children affected by generalized epilepsy and treated in vitro with myofibrillar, sarcoplasmic, and total protein fractions of meat and fish sources. 

## 2. Material and Methods

### 2.1. Patient Recruitment Procedure 

Children with generalized epilepsy (epilepsy group (EC): 8 males, 8 females; mean age, 32.4 ± 4.4 mo) and children without any clinical signs of diseases (control group (CC); 8 male, 8 female; mean age, 33.6 ± 5.8 mo) were recruited at the Complex Structure of Neuropsychiatry Childhood-Adolescence of Policlinico Riuniti (Foggia, Italy). The principal biochemical and anthropometric characteristics are reported in Table 1. The patient inclusion and exclusion criteria were determined with reference to the work of Albenzio et al. [19], and patients were treated with antiepileptic drugs (sodium valproate, 20 mg/kg/die). The study was approved by the local Ethics Committee on 22 May 2018, n°440/DS, the study was conducted according to the ethical principles of the Declaration of Helsinki, and written informed consent was obtained from the patients’ parents. 

### 2.2. Separation of Meat and Fish Proteins

Meat and fish samples were obtained from *longissimus thoracis* muscle for beef (BF) and lamb (LA); from *pectoralis* muscle for chicken (CH); and from dorsal white muscle for sole (*Solea solea*, SO), European hake (*Merluccius merluccius*, EH), and sea bass fish (*Dicentrarchus labrax*, *SB*), respectively. The separation of the two main meat and fish proteins fractions, myofibrillar (MYO) and sarcoplasmic (SA) proteins, was carried out following the method reported by della Malva et al. [20]. Briefly, meat samples were suspended in phosphate buffer (0.03 M, pH 7) containing a protease inhibitor cocktail (P2714, Sigma-Aldrich, St. Louis, MO, USA) and homogenized on ice for 2 min using an Ultra-Turrax T18 basic instrument (IKA, Wilmington, Germany). After centrifugation at 8000× *g* for 20 min at 4 °C, the supernatant representing the sarcoplasmic protein fraction was collected, and the pellet was subsequently treated with a non-denaturaing extraction method to extract myofibrillar proteins according to the protocol described by Hashimoto et al. [21], with some modifications as reported by della Malva et al. [20]. The non-denaturating extraction method was based on non-toxic and polluting agents; therefore, it resulted in a green solvent alternative method to mimic the effect of different protein fractions on peripheral mononuclear cells. Both myofibrillar and sarcoplasmic protein fractions were recovered into aqueous supernatant, then mixed (1:1) after protein concentration determination using a BCA protein assay kit (Thermo Fisher Scientific, Waltham, MA, USA) to obtain the total (TOT) protein fraction meat samples. On average, the TOT protein fraction concentrations for each sample were: 3571.76 µg/mL for lamb, 2743.08 µg/mL for European hake, 5082.06 µg/mL for chicken, 2749.01 µg/mL for sole, 2879.47 µg/mL for sea bass, and 5956.1 µg/mL for beef. All the meat and fish protein fractions (MYO, SAR, and TOT) were frozen at −80° for the lyophilization procedure.

### 2.3. SDS-PAGE Analysis of Fish and Meat Samples

Myofibrillar, sarcoplasmic, and total protein fractions from each meat source were resolved by SDS-polyacrylamide gel electrophoresis in an 8–18% gradient gel and analyzed according to method reported by della Malva et al. [20]. Gel was analyzed with Quantity One software Version 4.6.3 (Bio-Rad Laboratories, Hercules, CA, USA), and the respective protein molecular weight was obtained by comparison with precision plus protein standard-broad range (Bio-Rad Laboratories, Hercules, CA, USA).

### 2.4. Determination of Cytokine Profile from PBMCs Supernatants 

The levels of TNF-α, IL-10, IL-6, and IL-1β cytokines secreted by PBMC supernatants were determined after incubation with 100 μg/mL of different meat protein fractions (SA, MYO, and TOT) obtained from meat and fish species according to the protocol previously described by Albenzio et al. [16,19]. Peripheral blood was collected in heparinized tubes from EC and CC groups. The PBMC isolation procedure and protein fraction concentration used in in vitro the trial were chosen according to Albenzio et al. [19]. Briefly, PBMCs were isolated using Ficoll–Histopaque (Sigma Aldrich, Milan, Italy) density gradient centrifugation. All the proteins fractions were suspended into RPMI-1640 medium (Sigma Aldrich) supplemented with L-glutamine, penicillin/streptomycin, and 10% fetal bovine serum (Sigma Aldrich). A concentration of 1.5 × 10^5^ cells/mL was seeded into sterile 96- well microtiter plates and activated with the mitogen phytohemagglutinin (PHA, 10 μg/mL), 100 μL of each protein fraction was added to each well, and the plates were cultured for 5 days in 5% CO_2_ in a humidified incubator. PHA-activated PBMCs represented the positive control (PC), whereas PBMC with only culture medium was used as negative control (NC). All the treatments were tested in quadruplicate. PBMC viability after culturing time was determined using a trypan blue dye exclusion test and resulted in >90% viability. In vitro treatments were represented by PC, NC, SA-LA, MYO-LA, TOT-LA, SA-CH, MYO-CH, TOT-CH, SA-BF, MYO-BF, TOT-BF, SA-EH, MYO-EH, TOT-EH, SA-SO, MYO-SO, TOT-SO, SA-SB, MYO-SB, and TOT-SB. Cell-free supernatants were collected and immediately stored at −20 °C until cytokine profile determination using Luminex Multiplex assays (Thermo Fisher Scientific, Waltham, MA, USA).

### 2.5. Statistical Analysis

Data were tested for normal distribution using the Shapiro–Wilk test [22] and analyzed by ANOVA using the MIXED procedure of the SAS Institute (Cary, North Carolina, USA) [23]. The fixed effects were represented by meat proteins (sarcoplasmic fraction, myofibrillar fraction, and total protein fraction), meat sources (lamb, chicken, beef, European hake, sole, and sea bass), and their interactions. For cytokine response, patients were grouped according to Table 2, in which LL-EC represents the group of children with epilepsy with similar levels of cytokines to those of control children, ML-EC represents the group of children with epilepsy with cytokine levels at least 2-fold higher (medium levels) than those of control children, and HL-EC represents the group of children with epilepsy with cytokine levels at least 8-fold higher (high levels) than those of control children. To clarify the role of different cytokine groups in in vitro responses, additional fixed effects were tested, including classes (LL-EC, ML-EC, and HL-EC), the in vitro treatment (CS, CN, and all meat proteins fractions and species), and their interactions. When significant effects were found (at *p* < 0.05), Tukey post hoc test for multiple comparisons was used to identify significant differences between means. *p* = 0.10 was considered a tendency. 

## 3. Results 

### 3.1. Characterization of Proteomic Profile of Meat and Fish Extracts 

The proteomic profiles of meat and fish (Figure 1a and Figure 1b, respectively) sources revealed that the main myofibrillar proteins were myosin heavy chain, α-actinin, desmin, actin, troponin T, tropomyosin, myosin light chain 1, and myosin light chain 2. Sarcoplasmic proteins were mainly characterized by glycolytic enzymes, including glycogen phosphorylase b kinase, phosphoglucomutase, phosphoglucose isomerase, enolase, creatine kinase, glyceraldehyde phosphate dehydrogenase, phosphoglycerate mutase, and triosephosphate isomerase. However, all the species analyzed showed protein fragments in the molecular weight range of 180 to 110 KDa, mainly represented by the myosin heavy chain from 95 to 60 KDa, with the principal bands ascribed to α-actinin and phosphoglucomutase. In the lower part of the electrophoretogram, protein fragments derived from actinin, troponin T, glyceraldehyde, and myosin light-chain isoforms were detected.

### 3.2. Cytokine Pattern 

The cytokine pattern, in terms of TNF-α (Figure 2), IL-6 (Figure 3) and IL-10 (Figure 4), was not affected by different meat fractions and meat sources according to analysis of data within each class of cytokine response (LL-EC, ML-EC, and HL-EC). 

On the contrary, IL-1β in the HL-EC class showed a significant effect for the different protein fractions tested (*p* = 0.004), different meat sources (*p* = 0.001), and their interaction (*p* = 0.001). On average, MYO protein fractions had lower levels of IL-1β than SAR protein fractions (*p* = 0.003). Additionally, protein fractions obtained from LB, BF, and CH meat sources showed a higher level of IL-1β than protein fractions obtained from EH and SB fish samples, on average. The interaction between different protein fractions and meat sources demonstrated that MYO-EH had the lowest level of IL-1β (Figure 5). In order to facilitate the interpretation of results, Figure 2, Figure 3, Figure 4 and Figure 5 show the interaction between the different meat fractions and meat sources in cytokine classes LL-EC, ML-EC, and HL-EC. 

Data were further analyzed, including the effects of the three cytokine classes, the in vitro treatments, and their interaction. Cytokine classes were found to significantly affect the cytokine pattern (*p* < 0.001 for all cytokines). Furthermore, with respect to the effect of all the in vitro treatments including SC and NC on the three cytokine classes was found to be significant, on average, for TNF-α (*p* < 0.001), IL-10 (*p* = 0.01), IL-1β (*p* < 0.001), and as a tendency for IL-6 (*p* = 0.06) (Figure 6a–d). The interaction between the three cytokine classes and in vitro treatments was found to be significant for IL-10, IL-1β, and TNF-α (*p* < 0.001). To simplify data presentation, the effects of all in vitro treatments are presented in Figure 6. The IL-10 secretion of PC cells did not differ between MYO protein from LA and CH, and between SA protein from CH meat sources. However, MYO protein fractions from EH, SO, and BF sources; SA protein fractions from LA, EH, SO, and SB; and TOT proteins from all the meat sources showed significantly lower levels those of PC (Figure 6a). Regarding the level of IL-1β, MYO protein fractions from EH and TOT, as well as protein fractions from SB meat sources, were reduced as compared to PC and all other meat protein fractions and sources (Figure 6b). On the contrary, TNF-α secretion did not differ between in vitro treatments, showing the lowest levels in NC PBMC, as expected (Figure 6c). Similar levels of IL-6 secretion were observed in PC in MYO protein fractions from SB meat sources and in SA protein fractions from EH, CH, and BF meat sources. All other in vitro treatments resulted in IL-6 levels comparable to those of NC PBMC (Figure 6d).

## 4. Discussion

The role of inflammation in epileptic seizure is the subject of debate; however, it has been confirmed that an inflammatory state is originated from seizure-induced damage to the brain and from excessive muscular activity [24]. Cytokines are inflammatory mediators that regulate cell growth, activation, and differentiation, as well as inflammatory response [25]. An emerging area of research focuses on determining the extent of involvement of peripheral immune cells in the inflammatory pathology of epilepsy [26].

Previous studies have demonstrated that protein fractions from bovine, caprine, and ovine milk can modulate the secretion of cytokines in in vitro PMBC isolated from infants with generalized epilepsy [19]. The differences in cytokine responses are associated with the genetic polymorphisms of milk proteins; β-CN fractions from ovine and caprine milk were found to induce the highest levels of TNF-α, whereas αS2- CN fractions from bovine milk were found to produce the highest levels of IL-6 play a major role in production of IL-1β [27].

To the best of our knowledge, this is the first study in which the possible immunostimulatory or immunomodulatory role, in terms of cytokine pattern, of the main protein fractions extracted from different meat sources has been investigated in PBMCs of children with generalized epilepsy. 

A multitude of scientific evidence of the immune-related compounds involved in epilepsy demonstrates that PBMCs from epilepsy patients after an in vitro challenge with different stimulators produce increased amounts of inflammatory mediators of the IL-1 family (IL-6, IL-10, and IFN-α) compared to PBMCs of control patients [28,29,30,31,32,33]. Therefore, if the immune-related cells of patients with epilepsy, upon stimulation, react with a “hyper-reactive immune state”. Accordingly, in the present study, the high reactive immune state of epileptic patients was discovered in ML-EC and HL-EC classes, although not all epileptic patients responded to mitogen stimulation at the same level for each cytokine. More that 50% of epileptic patients had similar cytokine levels to those of control patients. For TNF-α, IL-6, IL-1β, and IL-10, at least 10% of epileptic patients was classified into the LL and HL classes. 

Cytokine pattern has a relevant role in epileptic disorders; therefore, the use of cytokine inhibitors has been studied as a possible treatment for epilepsy in animal studies. It was found that a recombinant IL-1 receptor antagonist significantly reduced seizure duration and recurrence [34,35,36], as well as incidence of status epilepticus in rodents [37]. A possible upregulation or downregulation of proinflammatory cytokines, such as IL-6, IL-1β, TNF-α; as well as anti-inflammatory cytokines, such as IL-4, IL-6, IL-10, IL-11, and IL-13, can be mediated by nutraceutical which defined an immune-boosting property of such foods [38]. Nutritional treatment with low levels of proinflammatory nutrients and high levels of nutrients with anti-inflammatory properties may be conceived against autoimmune and other chronic diseases. Red meat has proinflammatory properties, as a dietary source of arachidonic acid, which plays a role in the production of proinflammatory eicosanoids [39]. In the present study, the proinflammatory IL-1β in hyper-reactive immune-state patients (LL-EC) was decreased by MYO, SAR, and TOT protein fractions from European hake and sole fish meat sources. The reductive role of the pro-inflammatory IL-1β of myofibrillar protein fraction from fish meat sources was confirmed by averaging cytokine classes. This clearly demonstrated that myofibrillar fractions from fish sources can exert an inflammation-modulatory role in PBMCs from patients with epilepsy, such as a model of systemic response. On the contrary, PBMCs from epileptic patients and in the presence of bovine whey protein fractions, as well as caprine milk and its casein fractions, induced the highest levels of IL-1β in 60% and 80% of cases, respectively [27]. This finding is of relevance for the key role of the IL-1β system and contributes to epileptic seizures and the possible effect diet mediation. Recently, novel targets for pharmacological intervention based on anti-IL-1β treatments using small molecules or peptides have been experimentally investigated in clinical use for treatment of peripheral chronic inflammatory diseases [40]. In particular, the compromised blood–brain barrier (BBB) in epileptogenic brain tissue [41] suggests that brain penetration of systemically delivered drugs is preferential in diseased tissue with ongoing proinflammatory processes [42,43,44]. Therefore, increases in several inflammatory mediators, including the IL-1β and IL-1 receptor type 1 (IL-1R1), found in epileptogenic brain specimens in patients with drug-resistant epilepsy of different etiologies could be treated with IL-1-converting enzyme (ICE)/caspase 1 inhibitors and IL-1β receptor antagonists [40]. 

Kim and Cho [45] summarized a selection of nutritional suggestions that have proven beneficial in treating different types of epilepsy; fundamental molecular mechanisms, followed by an introduction to several functional nutrients that have the potential to manage seizure frequencies, were ascribed to molecules as omega-3 PUFA, vitamin D3, vitamin E, vitamin B6, vitamin C, and pyruvate. In this context, beneficial effects may be addressed by animal-derived food products naturally enriched with bioactive and functional nutrients by means of husbandry strategies such as animal feed supplements or natural pasture [46]. Moreover, the introduction of a balanced amount of meat in the diet of children with generalized epilepsy treated with nutritional intervention could provide an adequate number of essentials amino acids and fatty acids for growth and development and could help to control the activation of seizures mediated by severe deficiency of essential nutrients in the diet. 

In a study of high-fat-fed rats, fish proteins were found to exert an anti-inflammatory effect by reducing TNF-α and IL-6 expression in visceral adipose tissue. Moreover, in LPS-treated macrophages, not all fish protein sources exerted anti-inflammatory actions via the same mechanism, which depended mainly to the model tested, such as adipose tissue [47]. Growing interest relies on the role of fish protein and the beneficial actions of fish consumption in glucose metabolism and inflammation, even if the related specific mechanisms remain unknown. One possible explanation for the beneficial properties of fish protein may be the combination of high proportions of certain amino acids, such as arginine and taurine, with low levels of branched-chain amino acids found in fish meat [47]. Furthermore, the demonstrated bioactive properties of fish proteins include the control of plasma cholesterol, lipoproteins, and triglycerides [48], as well as angiotensin-converting enzyme activity [49]. 

IL-10 is an anti-inflammatory cytokine known as a human cytokine synthesis inhibitor factor (CSIF) [50]. IL-10 was found to limit inflammation in the brain, and it accounts for the inhibition of the synthesis of proinflammatory agents, such as interferon-(IFN)γ, IL-2, IL-3, TNF-α, and granulocyte macrophage colony-stimulating factor (GM-CSF) [51]. In a prenatally stressed rat model of febrile seizures, the potential therapeutic effect of quercetin was studied, demonstrating that quercetin increased IL-10 levels, restoring them to the basal level found in non-stressed rats with febrile seizures by suppressing proinflammatory cytokines and suggesting its neuroprotective effect [52]. However, few studies have dealt with the alteration of IL-10 in epileptic patients; moreover, data from animal experiments revealed an enhancement of IL-10 serum levels in neonatal seizures induced by hypoxic–ischemic encephalopathy [53]; therefore, it may be useful as an antiseizure agent. On the contrary, no changes in plasma IL-10 were observed in children with febrile seizures [54]. Accordingly, in epileptic patients 72 h post seizure attack, peripheral IL-10 was not found to be different as compared with controls [55]. Moreover, in a study on postictal and interictal phases in active epilepsy patients, neither postictal nor interictal plasma IL-10 changed, suggesting that IL-10 may not exert antiseizure action during this period [56]. Cultured PBMCs from infants with generalized epilepsy in the presence of protein fractions from different milk sources resulted in an immunological state hypoactivated, with low levels of IL-10 detected in supernatants [27]. In PBMCs from infants with cow milk allergy and in the presence of bovine and caprine milk fractions, a high amount of IL-10 was detected as compared to healthy infants [17]. In the present experiment, myofibrillar protein from EH, SO, and BF sources; sarcoplasmic protein fractions from LA, EH, SO, and SB sources; and TOT protein from all meat sources were found to have a lower level of IL-10 than positive controls, demonstrating a potential inhibitory role of some proteins in the release of IL-10, although no significant increment of IL-10 production was found any protein fractions in comparison to positive controls. This last concept can further explain the controversial contribution of IL-10 in the epilepsy. 

Protein fractions from meat and fish can be considered a valid therapeutic strategy to treat children with epilepsy by reducing the inflammation state. Different protein fractions from meat and fish sources proved to be effective in the modulation of cytokines in our ex vivo study, suggesting that beneficial nutrients may be delivered in the diet with therapeutic efficacy to reducing seizures or, in some cases, to replace drugs in epileptic children.

## 5. Conclusions

Data obtained the present study demonstrate that protein extracts from different meat sources have a moderate effect on cytokine patterns of infants suffering from generalized epilepsy. Particularly evident was the modulatory role on the proinflammatory IL-1β exerted by myofibrillar protein fractions extracted from fish sources. In addition, a slight anti-inflammatory effect on IL-10 production was demonstrated by myofibrillar and sarcoplasmic protein fractions from fish and meat sources, as well as all total protein fractions when analyzed in terms of averaged cytokine classes. Our findings present an opportunity to further validate nutritional interventions based on protein from animal sources to modulate the immunological cytokine pattern of infants with generalized epilepsy; however, further studies are needed to clarify the underlying specific molecular mechanism. 

## Figures and Tables

**Figure 1 nutrients-14-02243-f001:**
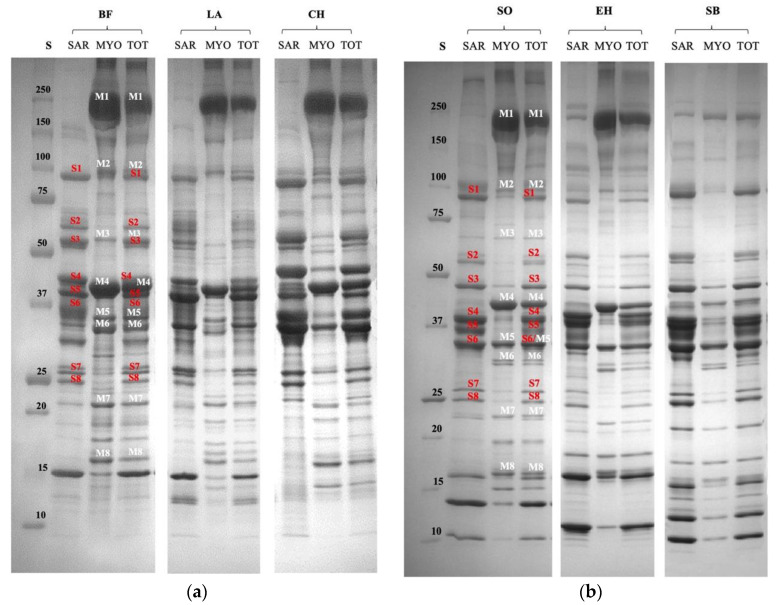
Gradient (8–18%) SDS-PAGE gel of myofibrillar (MYO), sarcoplasmic (SAR), and total (TOT) protein fractions extracted from beef (BF), lamb (LA), and chicken (CH) meat samples (**a**); and sole (SO), European hake (EH), and sea bass fish (SB) meat sources (**b**). M1 = myosin heavy chain; M2 = α-actinin; M3 = desmin; M4 = actin; M5 = troponin T; M6 = tropomyosin; M7 = myosin light chain 1; M8 = myosin light chain 2; S1 = glycogen phosphorylase b kinase; S2 = phosphoglucomutase; S3 = phosphoglucose isomerase; S4 = enolase; S5 = creatine kinase; S6 = glyceraldehyde phosphate dehydrogenase; S7 = phosphoglycerate mutase; S8 = triosephosphate isomerase.

**Figure 2 nutrients-14-02243-f002:**
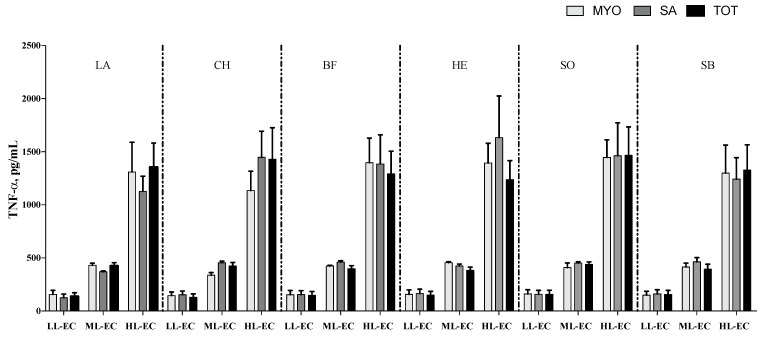
TNF-α secretion (means ± SE; pg/mL) in peripheral blood mononuclear cell supernatants of infants with generalized epilepsy treated in vitro with myofibrillar (MYO, light-grey bar), sarcoplasmic (SA, grey bar), and total protein (TOT, black bar) fractions from beef (BF), lamb (LA), chicken (CH), sole (SO), European hake (EH), and sea bass (SB) sources. LL-EC = group of children with epilepsy with similar levels of cytokines to those of control children; ML-EC = group of children with epilepsy with cytokine levels at least 2-fold higher (medium levels) than those of control children; HL-EC = group of children with epilepsy with cytokine levels at least 8-fold higher (high levels) than those of control children.

**Figure 3 nutrients-14-02243-f003:**
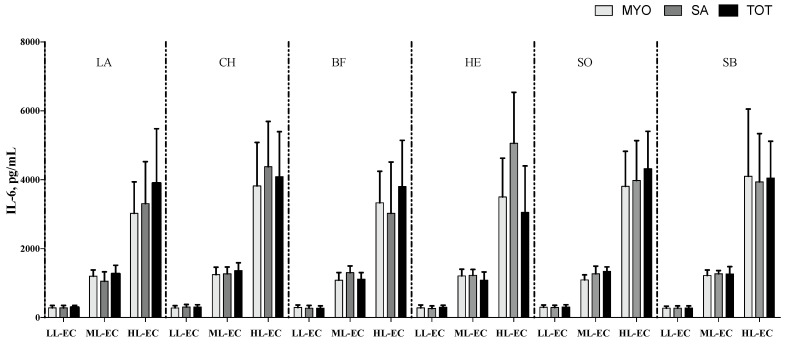
IL-6 secretion (means ± SE; pg/mL) in peripheral blood mononuclear cell supernatants of infants with generalized epilepsy treated in vitro with myofibrillar (MYO, light-grey bar), sarcoplasmic (SA, grey bar), and total protein (TOT, black bar) fractions from beef (BF), lamb (LA), chicken (CH), sole (SO), European hake (EH), and sea bass (SB) sources. LL-EC = group of children with epilepsy with similar levels of cytokines to those of control children; ML-EC = group of children with epilepsy with cytokine levels at least 2-fold higher (medium levels) than those of control children; HL-EC = group of children with epilepsy with cytokine levels at least 8-fold higher (high levels) than those of control children.

**Figure 4 nutrients-14-02243-f004:**
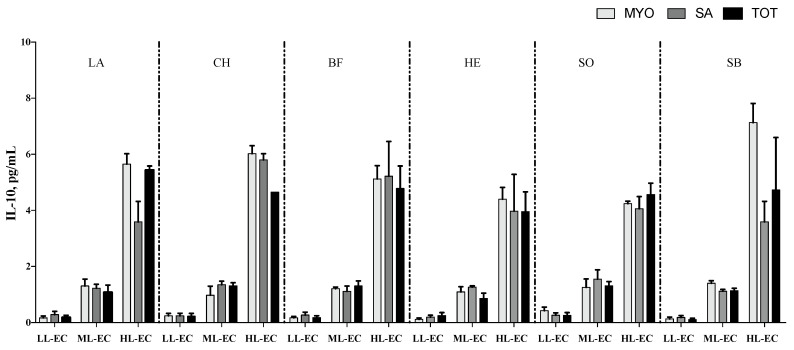
IL-10 secretion (means ± SE; pg/mL) in peripheral blood mononuclear cell supernatants of infants with generalized epilepsy treated in vitro with myofibrillar (MYO, light-grey bar), sarcoplasmic (SA, grey bar), and total protein (TOT, black bar) fractions from beef (BF), lamb (LA), chicken (CH), sole (SO), European hake (EH), and sea bass (SB) sources. LL-EC = group of children with epilepsy with similar levels of cytokines to those of control children; ML-EC = group of children with epilepsy with cytokine levels at least 2-fold higher (medium levels) than those of control children; HL-EC = group of children with epilepsy with cytokine levels at least 8-fold higher (high levels) than those of control children.

**Figure 5 nutrients-14-02243-f005:**
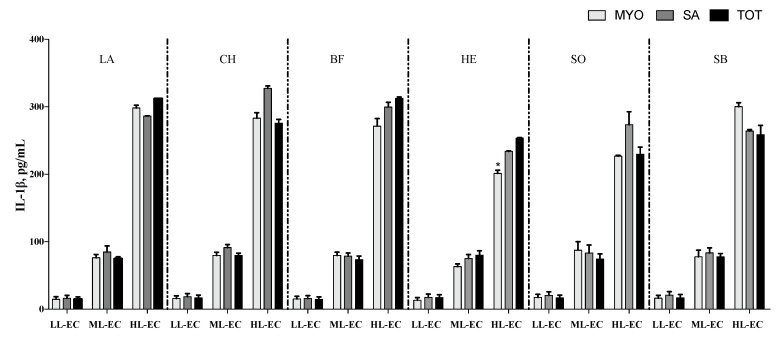
IL-1β secretion (means ± SE; pg/mL) in peripheral blood mononuclear cell supernatants of infants with generalized epilepsy treated in vitro with myofibrillar (MYO, light-grey bar), sarcoplasmic (SA, grey bar), and total protein (TOT, black bar) fractions from beef (BF), lamb (LA), chicken (CH), sole (SO), European hake (EH), and sea bass (SB) sources. LL-EC = group of children with epilepsy with similar levels of cytokines to those of control children; ML-EC = group of children with epilepsy with cytokine levels at least 2-fold higher (medium levels) than those of control children; HL-EC = group of children with epilepsy with cytokine levels at least 8-fold higher (high levels) than those of control children. * *p* < 0.05.

**Figure 6 nutrients-14-02243-f006:**
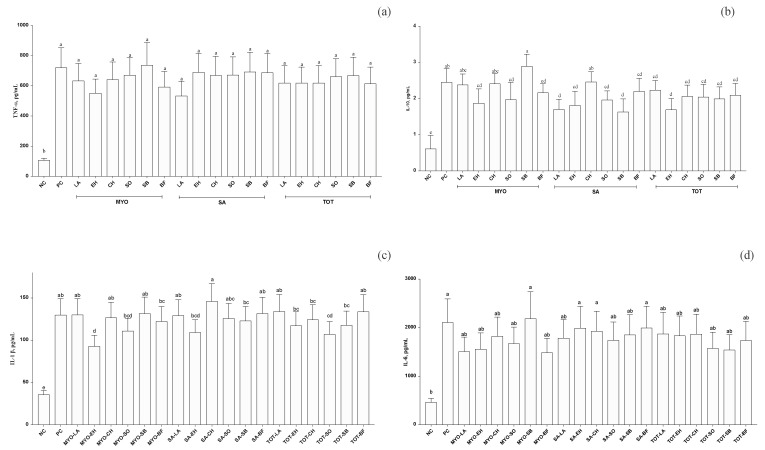
Cytokine pattern in terms of (**a**) TNF-α, (**b**) IL-10, (**c**) IL-1β, and (**d**) IL-6 (means ± SE; pg/mL) averaged among different classes (LL-EC = group of children with epilepsy with similar levels of cytokines to those of control children; ML-EC = group of children with epilepsy with cytokine levels at least 2-fold higher (medium levels) than those of control children; HL-EC = group of children with epilepsy with cytokine levels at least 8-fold higher (high levels) than those of control children) and based on in vitro treatments (NC = negative control, PC = positive control, SA-LA = sarcoplasmic protein fractions from lamb meat, MYO-LA = myofibrillar protein fraction from lamb meat, TOT-LA = total protein fractions from lamb meat, SA-CH = sarcoplasmic protein fractions from chicken meat, MYO-CH = myofibrillar protein fractions from chicken meat, TOT-CH = total protein fractions from chicken meat, SA-BF = sarcoplasmic protein fractions from beef, MYO-BF = myofibrillar protein fractions from beef, TOT-BF = total protein fractions from beef, SA-EH = sarcoplasmic protein fractions from European hake fish meat, MYO-EH = myofibrillar protein fractions from European hake fish meat, TOT-EH = total protein fractions from European hake fish meat, SA-SO = sarcoplasmic protein fractions from sole fish meat, MYO-SO = myofibrillar protein fractions from sole fish meat, TOT-SO = total protein fractions from sole fish meat, SA-SB = sarcoplasmic protein fractions from sea bass, MYO-SB = myofibrillar protein fractions from sea bass, and TOT-SB = total protein fractions from sea bass. ^a–d^ Means with different letters are significantly different, *p* < 0.05.

**Table 1 nutrients-14-02243-t001:** Main anthropometric and biochemical characteristics of children with generalized epilepsy (EC) and control children (CC) recruited for the study.

	EC	CC	*p*-Value
M/F	8/8	8/8	ns
AGE (MO)	32.4 ± 4.4	33.6 ± 5.8	ns
PERCENTILE	25°	25°	ns
GLYCEMIA (MG/DL)	75.5 ± 2.8	79.8 ± 3.6	ns
HEMOGLOBIN (G/DL)	12.9 ± 2.3	11.9 ±1.2	ns
TOTAL CHOLESTEROL (MG/DL)	110 ± 5.6	115 ± 3.4	ns
TRIGLYCERIDES (MG/DL)	90.6 ± 6.1	109.7 ± 4.3	ns
ALBUMIN (G/DL)	3.5 ± 0.8	4 ± 2.1	ns
CRP (MG/DL)	<0.6	<0.6	ns

NS = not significant, M/F = Males/Females, CRP = C-Reactive Protein, *p*-value > 0.05.

**Table 2 nutrients-14-02243-t002:** Percentage distribution of children with epilepsy grouped according to levels of different cytokines in peripheral blood mononuclear cell supernatants.

	Class ^1^		
Cytokines	LL-EC	ML-EC	HL-EC
TNF-α	60	20	20
IL-6	60	20	20
IL-1β	60	30	10
IL-10	70	20	10

^1^ LL-EC = group of children with epilepsy with similar levels of cytokines to those of control children; ML-EC = group of children with epilepsy with cytokine levels at least 2-fold higher (medium levels) than those of control children; HL-EC = group of children with epilepsy with cytokine levels at least 8-fold higher (high levels) than those of control children. TNF = Tumor necrosis factor, IL= Interleukin.

## Data Availability

All relevant data are presented within the manuscript.

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
