# Peer review of "Cytokine Pattern of Peripheral Blood Mononuclear Cells Isolated from Children Affected by Generalized Epilepsy Treated with Different Protein Fractions of Meat Sources"

_nutrients, 2022, doi:10.3390/nu14112243_

Round 1
Reviewer 1 Report
In this study, Cilberti et al analyzed the cytokine pattern in peripheral blood mononuclear cells (PBMCs) supernatants isolated from children affected by generalized epilepsy. In addition, the authors examined in vitro effects of different protein fractions extracted from different sources on cytokine profile, suggesting the potential immunomodulatory role in children with generalized epilepsy. Overall, this is well written manuscript and reports out interesting findings that can be considered to develop a nutritional strategy for children with such neurological disorder. I have following comments:
1) Why did authors choose PBMCs to analyze cytokine profile in this study? This should be elaborated in Introduction as well as discussion section.
2) The races of subjects involved in this study are unclear. How do authors explain the effects of different genetic background on cytokine profile and severity of epilepsy in these children? This should be further elaborated.
3) It would be interesting if the authors could use glial cell lines with same genetic background and treat with protein fractions from different sources and examine the cytokine production. This may provide consensus data eliminating the impacts of genetic background between the samples.
Author Response
Reviewer 1
In this study, Cilberti et al analyzed the cytokine pattern in peripheral blood mononuclear cells (PBMCs) supernatants isolated from children affected by generalized epilepsy. In addition, the authors examined in vitro effects of different protein fractions extracted from different sources on cytokine profile, suggesting the potential immunomodulatory role in children with generalized epilepsy. Overall, this is well written manuscript and reports out interesting findings that can be considered to develop a nutritional strategy for children with such neurological disorder. I have following comments:
1) Why did authors choose PBMCs to analyze cytokine profile in this study? This should be elaborated in Introduction as well as discussion section.
AU: Thanks for your question, we used PBMC to analyze cytokines’ profile also in our previous papers (Albenzio et al. 2016 and 2018) to understand the relation between different milk protein fractions derived from different species in children with generalized epilepsy at peripheral level which can be interconnected to the passage of cytokines through the blood brain barrier and involved in the epilepsy seizures as a key mechanism of the involvement of neuroinflammation, as a chronic brain disorder, that characterized the pathophysiology of epilepsy. Indeed, recurrent seizures of epilepsy have been associated with elevated levels of immune mediators that play a pivotal role in triggering them. Neurons, glia, and endothelial cells of the blood-brain barrier take part in such inflammatory processes by expressing receptors of associated mediators through autocrine and paracrine stimulation of intracellular signaling pathways. In this milieu, elevated cytokine levels in serum and brain tissue have been reported in patients with an epileptic profile. Among those, interleukin (IL)-1β, IL-6, and tumor necrosis factor-alpha (TNF-α) are the proinflammatory cytokines mostly associated, in literature, with the pathogenesis of epilepsies (Kamali, A. et al., 2021. The potential role of pro-inflammatory and anti-inflammatory cytokines in epilepsy pathogenesis. https://doi.org/10.2174/1871530320999201116200940).
Based on previous statements, the central role on PBMC study is related to the recent research on the pathogenesis of epilepsy which has started to evaluate the mechanisms mediating peripheral-to-central cell infiltration in both human and mouse models (Xu et al., 2018). Therefore, an emerging area of research focuses on determining the extent of involvement of peripheral immune cells in the inflammatory pathology of epilepsy (Yamanaka et al. 2021, Interleukin-1β in peripheral monocytes is associated with seizure frequency in pediatric drug-resistant epilepsy. https://doi.org/10.1016/j.jneuroim.2021.577475).
The introduction and discussion section have been implemented with PBMC analysis on cytokine profile, as requested. However, in the discussion section have been already reported “numerous scientific evidence on the immune related compounds involved in the epilepsy demonstrated that PBMCs from epilepsy patients after an in vitro challenge with different stimulators, produced increased amounts of inflammatory mediators of the IL-1 family, IL-6, IL-10, and IFN-α compared to PBMCs of controls [26; 27; 28; 29; 30; 31]. Therefore, if the immune related cells are upon stimulation, patients with epilepsy reacts with an “hyper-reactive immune state”. Accordingly, in the present study the high re-active immune state of epileptic patients was discovered in ML-EC and HL-EC classes, even if not all the epileptic patients responded to the mitogen stimulation at the same level for each cytokine. Mainly more that 50% of epileptic patients had similar cytokines’ level of control patients. For TNF-α, IL-6, IL-1β and IL-10 at least the 10% of epileptic patients was classified into LL and HL classes.”
2) The races of subjects involved in this study are unclear. How do authors explain the effects of different genetic background on cytokine profile and severity of epilepsy in these children? This should be further elaborated.
AU: We thank the reviewer for the important suggestion and take this as a starting point for further investigations on these patients of ours. As reported in the study, we investigated whether PMBC treatment with meat proteins can influence cytokine production. It is certainly interesting to also characterize the genetic profile of our patients from whom we extracted PMBC to understand what the genetic involvement is. Let's consider this tip for further study.
3) It would be interesting if the authors could use glial cell lines with same genetic background and treat with protein fractions from different sources and examine the cytokine production. This may provide consensus data eliminating the impacts of genetic background between the samples.
AU: We agree with the reviewer, the use of glial cell lines with same genetic background is a useful suggestion for further studies in order to avoid the different genetic background of patients under studies. However, our objective was to understand the potential contribute of protein from meat sources in the inflammatory responses of children with generalized epilepsy as a measure of predictive inflammatory state mediated by dietary proteins. Present data demonstrated that the different genetic background can be on the basis of different inflammatory responses, therefore, it could be interesting evaluate the inflammatory potential of nutrients in relation to the responsiveness of each individual and formulate specific diet for single case.
Reviewer 2 Report
In the manuscript “Cytokine pattern of peripheral blood mononuclear cells isolated from children affected by generalized epilepsy treated with different protein fractions of meat sources” authors present scientific evidence about the effect of diet on cytokines pattern produced by cells of children with epilepsy. The topic is interesting; however, the authors should substantially improve the manuscript, the following points could be addressed by authors.
- What was the rationale for n=8? Was n=8 or n=16? What is the reason for including in this study 8 male and 8 female? Is there a difference in the cytokines pattern according to sex?
- The authors should provide more information about the health of children included in the study. According with their reference, children with history of clinical seizures within the last 3 days before blood drawing were excluded. However, how many seizures by period (maybe a month), presented the children included in the study. Were the children treated with antiepileptic drugs?
- What was the rationale behind choosing 100 µg/ml meat proteins’ fractions? Was it based on previously established protocols? If so, provide references.
- My main concern is that the result section has a several errors that difficult to evaluate the significance of the findings that authors claim. This section should be carefully revised and edited appropriately. The following points should be revised:
- The description of results for figures 3, 4 and 5 do not correlate with the figures, so it is not possible to evaluate the relevance of these findings.
- Figures 2, 3, 4 and 5 do not include the legend for color marks.
- The asterisks that represent the statistical difference are moved in figure 4.
- What’s the meaning of letters on errors bar in figure 6? The description is not in figure legend. Please, describe appropriately.
- The discussion section is very long and some information discussed about published reports without highlighting the results of this work.
Author Response
Reviewer 2
In the manuscript “Cytokine pattern of peripheral blood mononuclear cells isolated from children affected by generalized epilepsy treated with different protein fractions of meat sources” authors present scientific evidence about the effect of diet on cytokines pattern produced by cells of children with epilepsy. The topic is interesting; however, the authors should substantially improve the manuscript, the following points could be addressed by authors.
- What was the rationale for n=8? Was n=8 or n=16? What is the reason for including in this study 8 male and 8 female? Is there a difference in the cytokines pattern according to sex?
AU: In this study, were enrolled 16 subjects with epilepsy and 16 controls (we corrected in the abstract, there is a mistake). 8 males and 8 females were enrolled because, as reported in the literature, there could be a different cytokine expression linked to sex, especially for the activation of the immune system in response to pathogens. However, we found no difference between males and females [Everhardt Queen A, Moerdyk-Schauwecker M, McKee LM, Leamy LJ, Huet YM. Differential Expression of Inflammatory Cytokines and Stress Genes in Male and Female Mice in Response to a Lipopolysaccharide Challenge. PLoS One. 2016 Apr 27;11(4):e0152289. doi: 10.1371/journal.pone.0152289. PMID: 27120355; PMCID: PMC4847773].
- The authors should provide more information about the health of children included in the study. According with their reference, children with history of clinical seizures within the last 3 days before blood drawing were excluded. However, how many seizures by period (maybe a month), presented the children included in the study. Were the children treated with antiepileptic drugs?
AU: Exclusion criteria were occurrence of clinical seizures within the last 3 d before blood drawing, malignant tumor, concomitant inflammatory disease, severe neurological or neuroimmunological disease (i.e., stroke, cerebral hemorrhage, encephalitis, meningitis), immunosuppressive or immunomodulatory treatment during the last 6 mo, surgery or significant trauma within the last 2 wk, hepatic or renal insufficiency, or severe psychiatric disease. Moreover, the children enrolled were under antiepileptic drugs treatment, treated with sodium valproate, 20 mg/kg/die, which was added in the section 2.1. Furthermore, in the practice all the children enrolled in the study were free from seizure events in the last month before the blood sample collection.
- What was the rationale behind choosing 100 µg/ml meat proteins’ fractions? Was it based on previously established protocols? If so, provide references.
AU: References to previous protocols have been added in the section.
My main concern is that the result section has a several errors that difficult to evaluate the significance of the findings that authors claim. This section should be carefully revised and edited appropriately. The following points should be revised:
- The description of results for figures 3, 4 and 5 do not correlate with the figures, so it is not possible to evaluate the relevance of these findings.
AU: Results description for figure 3, 4, and 5 have been improved in order to clarify the relevance of the findings.
- Figures 2, 3, 4 and 5 do not include the legend for color marks.
AU: The legend for color marks has been included as requested by reviewer.
- The asterisks that represent the statistical difference are moved in figure 4.
AU: The asterisk of the Figure 4 was placed correctly.
- What’s the meaning of letters on errors bar in figure 6? The description is not in figure legend. Please, describe appropriately.
AU: The description on the meaning of letters on errors bar in figure 6 has been added, accordingly.
- The discussion section is very long and some information discussed about published reports without highlighting the results of this work.
AU: We thank the reviewer for the observation aimed to improve the overall quality of the discussion. According to the suggestion, the discussion of the relationship between cytokines and oxidative stress biomarkers has been deleted, because of the latter indexes were not the objective of the present paper.
Round 2
Reviewer 1 Report
The authors have made moderate changes in the manuscript.
Author Response
Thank you for your revision.
Reviewer 2 Report
My main criticism of the manuscript “Cytokine pattern of peripheral blood mononuclear cells isolated from children affected by generalized epilepsy treated with different protein fractions of meat sources” is that the authors have not reviewed carefully the presentation of their findings in the results section. This point was of my main concern in the revision round 1.
1. In the subsection 3.2 the authors claim that results of IL-10 secretion is in (Figure 3), IL-6 is in (Figure 4) and IL-1β is in (Figure 5). However, in the figures legends Figure 3 is IL-6 secretion, Figure 4 is IL-1β secretion and Figure 5 is IL-10 secretion. Which ones are the correct results?
2. Since the children included in this study were treated with sodium valproate, authors may discuss if this drug could influence their results obtained since it is known that valproate modulates the immune response and downregulated the macrophage responses through
the inhibition of the proinflammatory cytokines TNF-α, IL-1β, IL-6, IL-18, IL-12, and IFN-γ (DOI 10.1155/2019/9678098, doi: 10.1097/00004850-200511000-00002).
Author Response
Reviewer 2
My main criticism of the manuscript “Cytokine pattern of peripheral blood mononuclear cells isolated from children affected by generalized epilepsy treated with different protein fractions of meat sources” is that the authors have not reviewed carefully the presentation of their findings in the results section. This point was of my main concern in the revision round 1.
AU: Regarding results presentation, as we described in the statistical analysis, we performed two different types of analyses based on the cytokines classes. From Figure 2 to Figure 4 we presented cytokine pattern result based on analysis made separately from each classis; however, in order to facilitate data reading, all the classes were depicted in the same Figures. Only for IL-1β were found a significant effect for the different proteins’ fractions tested (P=0.004), different meat sources (P=0.001) and their interaction (P=0.001), as described in the results section. Differently from those results, cytokine pattern was analyzed on average of all cytokines classes in order to understand the effect of the in vitro treatments on the cytokines levels and their interaction respectively, as reported in the Figure 6 (a, b, c, and d)
- In the subsection 3.2 the authors claim that results of IL-10 secretion is in (Figure 3), IL-6 is in (Figure 4) and IL-1β is in (Figure 5). However, in the figures legends Figure 3 is IL-6 secretion, Figure 4 is IL-1β secretion and Figure 5 is IL-10 secretion. Which ones are the correct results?
AU: The Figure numbers has been correctly assigned to the ILs description; we are sorry for this mistake.
- Since the children included in this study were treated with sodium valproate, authors may discuss if this drug could influence their results obtained since it is known that valproate modulates the immune response and downregulated the macrophage responses through the inhibition of the proinflammatory cytokines TNF-α, IL-1β, IL-6, IL-18, IL-12, and IFN-γ (DOI 10.1155/2019/9678098, doi: 10.1097/00004850-200511000-00002).
AU: We thank the reviewer for this observation and for the suggested articles. The recruiting conditions of the epileptic patients was detailed as previously requested by the reviewer, in particular children received the drug treatment according to the specialist medical assessment. We agree with the reviewer that VPA administration, has an impact on the immune response, beyond specifically controlling epileptic events. Recently, literature reported that VPA has a pro-inflammatory effect in vitro and in vivo studies [Ismail FS, Corvace F, Faustmann PM, Faustmann TJ. Pharmacological Investigations in Glia Culture Model of Inflammation. Front Cell Neurosci. 2021 Dec 16;15:805755. doi: 10.3389/fncel.2021.805755. PMID: 34975415; PMCID: PMC8716582.; Błaszczyk B, Miziak B, Pluta R, Czuczwar SJ. Epilepsy in Pregnancy-Management Principles and Focus on Valproate. Int J Mol Sci. 2022 Jan 25;23(3):1369. doi: 10.3390/ijms23031369. PMID: 35163292; PMCID: PMC8836209.]. In this study, we reported that meat proteins are able to regulate the inflammatory response, so although there may be an influence of the drug, this does not however exclude the ability of the nutritional intervention to regulate the inflammatory response.